Corrected: Author Correction

# Repression of phagocytosis by human CD33 is not conserved with mouse CD33

Abhishek Bhattacherjee[1], Emily Rodrigues [1], Jaesoo Jung[1], Matthew Luzentales-Simpson[2], Jhon R. Enterina [2], Danny Galleguillos [3], Chris D. St. Laurent[1], Maryam Nakhaei-Nejad[4], Felix F. Fuchsberger[5], Laura Streith[2], Qian Wang[3], Norihito Kawasaki[6], Shiteng Duan[6], Arjun Bains[1], James C. Paulson [6], Christoph Rademacher [5], Fabrizio Giuliani[4], Simonetta Sipione[3] & Matthew S. Macauley [1,2]*

CD33 is an immunomodulatory receptor linked to Alzheimer's disease (AD) susceptibility via regulation of phagocytosis in microglia. Divergent features between human CD33 (hCD33) and murine CD33 (mCD33) include a unique transmembrane lysine in mCD33 and cytoplasmic tyrosine in hCD33. The functional consequences of these differences in restraining phagocytosis remains poorly understood. Using a new αmCD33 monoclonal antibody, we show that mCD33 is expressed at high levels on neutrophils and low levels on microglia. Notably, cell surface expression of mCD33 is entirely dependent on Dap12 due to an interaction with the transmembrane lysine in mCD33. In RAW264.7 cultured macrophages, BV-2 cultured microglia, primary neonatal and adult microglia, uptake of cargo — including aggregated $A\beta_{1-42}$ — is not altered upon genetic ablation of mCD33. Alternatively, deletion of hCD33 in monocytic cell lines increased cargo uptake. Moreover, transgenic mice expressing hCD33 in the microglial cell lineage showed repressed cargo uptake in primary microglia. Therefore, mCD33 and hCD33 have divergent roles in regulating phagocytosis, highlighting the importance of studying hCD33 in AD susceptibility.

[1] Department of Chemistry, University of Alberta, Alberta, Canada. [2] Department of Medical Microbiology and Immunology, University of Alberta, Alberta, Canada. [3] Department of Pharmacology, University of Alberta, Alberta, Canada. [4] Department of Medicine, University of Alberta, Alberta, Canada. [5] Department of Biomolecular Systems, Max Planck Institute of Colloids and Interfaces, Potsdam, Germany. [6] Department of Molecular Medicine, The Scripps Research Institute, La Jolla, CA 92037, USA. *email: macauley@ualberta.ca

Sialic acid-binding immunoglobulin-type lectins (Siglecs) are a family of immunomodulatory receptors with many ascribed roles in controlling immune cell function in health and disease[1]. In humans, 14 members of this family are differentially expressed on cells of the hemopoietic lineage, while nine family members exist in mice, reflecting the rapidly evolving nature of Siglecs[2]. Siglecs can be further subdivided into a conserved CD22-related subfamily and a divergent CD33-related subfamily[3–5]. Siglecs can act as either positive or negative regulators of immune cell signaling, which is primarily dictated by two residues. The first of these key residues is one or more tyrosines in the cytoplasmic tail that are contained within an immunoreceptor tyrosine-based inhibitory motif (ITIM). Under the appropriate physiological circumstance, these tyrosines can be phosphorylated by a Src family kinase and recruit a protein tyrosine phosphatase to dampen immune cell signaling. A second mutually exclusive key residue is a positively charged lysine or arginine in the transmembrane region in certain Siglec family members, which serves to pair Siglecs with an aspartic acid-containing adaptor protein, such as Dap12. These adaptor proteins have immunoreceptor tyrosine-based activatory motifs (ITAMs)[6,7], suggesting that Siglecs that pair with Dap12 have the potential to be activatory. Indeed, Siglec-H[8] and Siglec-15[9] in mouse, as well as Siglec-14[10], −15[11], and −16[12] in human contain this positively charged transmembrane residue and have ascribed activatory roles[13].

As has been noted in previous reviews, mouse CD33 (mCD33) has a transmembrane lysine residue and lacks a *bona fide* ITIM[2,3]. Conversely, human CD33 (hCD33) does not contain this transmembrane lysine and does contain a functional ITIM[14], although the physiological circumstances in which this ITIM is phosphorylated remains unclear. Little is known about the functional impact these key differences have on the function of mCD33 and hCD33 since the functional role(s) for CD33 (from mouse and man) in regulating immune cells has not been as easy to elucidate as other Siglec family members. Indeed, no significant phenotype was observed in CD33 knockout mice at the cellular or organismal level[15]. More recently, it was shown that mice reconstituted with hCD33$^{-/-}$ hematopoietic stem cells are healthy and show no significant alterations in immune cell function compared with their hCD33$^{+/+}$ counterparts[16].

A growing body of evidence implicates CD33 in controlling microglial cell function in the brain[17]. Genome-wide association studies revealed that a single nucleotide polymorphism (SNP) within the CD33 gene correlates with Alzheimer's disease (AD) susceptibility[18–20]. The common CD33 risk allele (rs12459419C) contains a cytidine near the start of exon 2, while the less common CD33 protective allele (rs12459419T) has a thymidine. Individuals with even one copy of the CD33 rare allele are statistically less likely to develop AD, with two copies being more protective[21]. At the amino acid level, the difference between C and T results in an alanine to valine alteration, but this position is contained within the signal sequence and, thus, not present in the fully mature protein. This SNP was, however, found to modulate alternative mRNA splicing[22,23]; exon 2 skipping increases substantially in the CD33 protective allele, leading to production of a short isoform previously described as hCD33m[24,25]. Accordingly, hCD33m lacks its N-terminal sialic acid-binding domain compared with the longer isoform known as hCD33M. Using peripheral blood monocytes[21,26], or monocyte-derived microglia[27], it was demonstrated that the copy number of the CD33 protective allele correlates with decreased expression of hCD33M and an increased ability to phagocytose cargo such as fluorescent dextran particles and amyloid-β peptide. It is noteworthy that the hCD33m isoform, lacking its glycan-binding domain, appears to be unique to humans[28].

Most phagocytic receptors drive Syk-dependent cellular signaling to promote cytoskeletal rearrangement[29–31]. Therefore, it is highly relevant that inhibitory-type Siglecs can efficiently inhibit Syk-driven cellular signaling through recruitment of phosphatases, such as SHP-1 and SHP-2, that directly dephosphorylate Syk and proximal signaling components[1]. One possible mechanism for the correlation between AD susceptibility and CD33 alleles is that the common CD33 risk allele, which preferentially gives rise to the long isoform (hCD33M), restrains phagocytosis in brain microglia, leading to the slow accumulation of aggregated amyloid-β peptides and thereby increasing the probability of neurodegenerative plaque deposition. It is noteworthy that other models can be envisioned whereby increased expression of the short isoform (hCD33m) has a yet undiscovered protective function. Indeed, a recent metagenomics analysis supporting a possible gain-of-function for hCD33m was proposed[17], and hCD33m was recently reported to inefficiently reach the cell surface[32]. Support for the loss-of-function model comes from studies wherein mCD33$^{-/-}$ mice crossed with the APP/PS1 or 5XFAD mouse model of Aβ plaque accumulation had decreased plaque accumulation compared with their mCD33-expressing counterparts[33,34]. Likewise, cultured microglia from WT and mCD33$^{-/-}$ neonatal mice were reported to have a differential ability to phagocytose fluorescent Aβ$_{1-42}$, with mCD33$^{-/-}$ microglia having a significantly increased phagocytic capacity.

In the context of the clear differences between mCD33 and hCD33 at certain key functional residues, we were interested in how mCD33 would be able to play an inhibitory role. To begin addressing this question, we developed a monoclonal antibody for mCD33 since previous studies relied on polyclonal antibodies[15,33]. We demonstrate, for the first time to our knowledge, that mCD33 expression on the cell surface requires Dap12. Through genetic manipulation of mCD33 and hCD33 expression in cultured cells and mouse microglia, we address the role of these two proteins in restraining phagocytosis and find strong support for hCD33M in restraining phagocytosis, which is not recapitulated by mCD33. Key to these findings is development of a new transgenic mouse model expressing hCD33 in the microglial cell lineage, which will provide an appropriate mouse model to probe the link between hCD33 and plaque accumulation.

## Results

**Expression analysis of mCD33 on immune cell subsets.** With the growing connection between CD33 and AD, and the utility of mouse models of AD for studying mechanisms of disease pathogenesis[35,36], we felt that clarifying the expression levels of mCD33 on immune cell subsets—including primary microglia—would be useful. Previous work analyzing the expression of mCD33 on immune cells relied on polyclonal antibodies[15,33], therefore, we established a monoclonal antibody toward mCD33 using a similar approach as we described previously for generation of an αSiglec-F monoclonal antibody[37]. A rat IgG1 monoclonal antibody (clone 9A11) was developed for detection of cell surface mCD33 by flow cytometry (Supplementary Fig. 1). Using this antibody, the expression levels of mCD33 were assessed on immune cells from WT and mCD33$^{-/-}$ mice (Fig. 1). In the spleen, we find the highest expression of mCD33 on neutrophils (CD11b$^+$ Ly-6G$^+$ Ly-6C$^-$ Cx$_3$cre1$^-$ F4/80$^-$; Fig. 1a). Low expression was found on plasmacytoid dendric cells (pDCs; Siglec-H$^+$ CD11c$^+$ B220$^+$ Ly-6C$^+$ CD11b$^-$), Fig. 1b), monocytes (CD11b$^+$ Ly-6C$^+$ Cx$_3$cr1$^+$ F4/80$^+$; Fig. 1c), and dendritic cells (DCs; CD19$^-$ CD11b$^{low}$ CD11c$^+$ MHCII$^{++}$; Fig. 1d). Negligible expression levels were found on natural killer cells (NK1.1$^+$ CD19$^-$ CD3$^-$ CD11b$^-$ Ly-6C$^-$; Fig. 1e), B cells (CD19$^+$ B220$^+$;

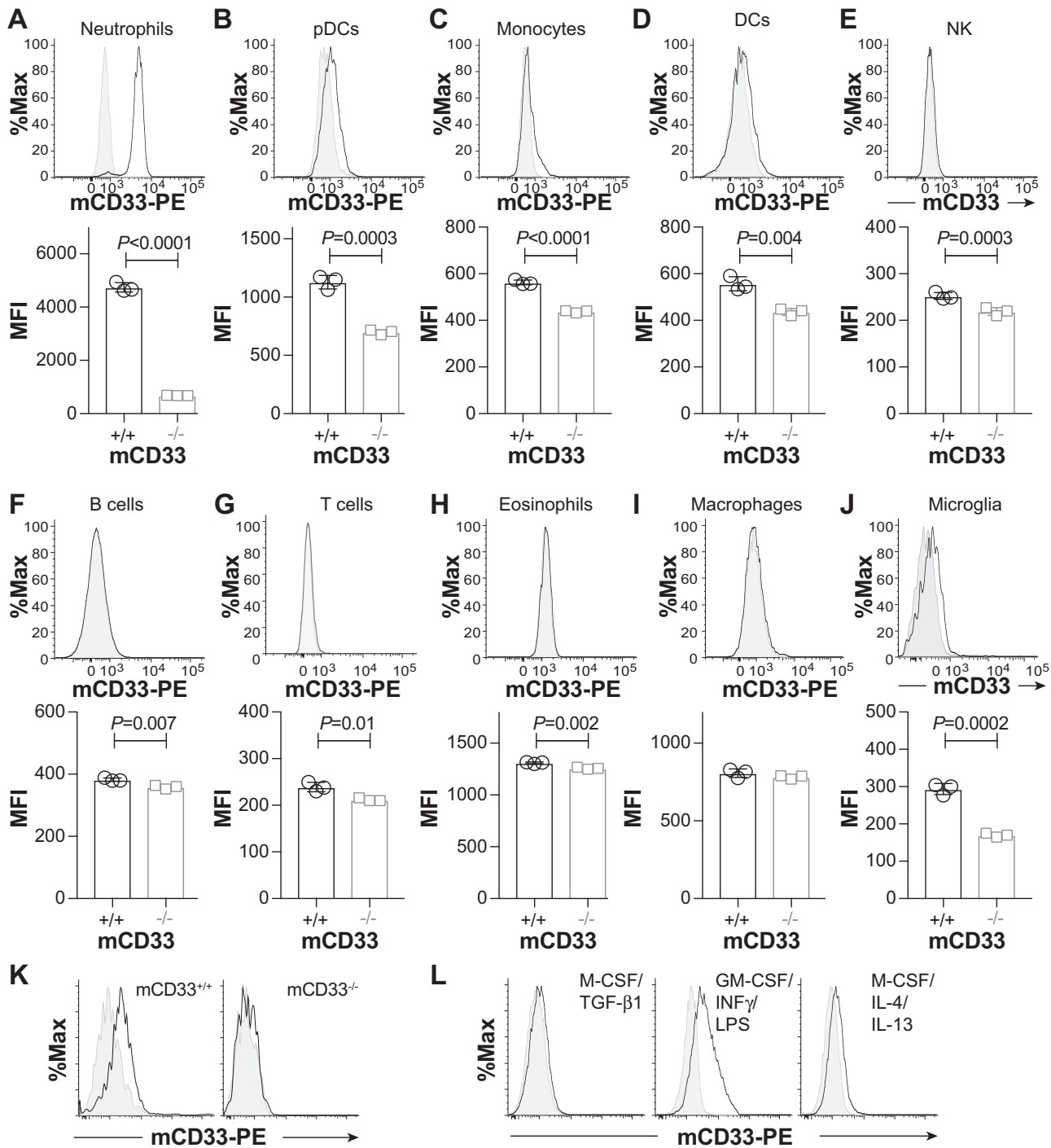

**Fig. 1** Expression profile of mCD33 on immune cell subsets. **a–i** A monoclonal rat IgG1 anti-mouse antibody was used to probe the expression of mCD33 on splenic leukocytes by flow cytometry. For each cell type, a representative histogram (gray: isotype control; black: αmCD33) is shown on the top and a bar graph of the median fluorescence intensity (MFI) for three individual WT (mCD33+/+) and CD33-deficient (mCD33−/−) mice. **j** Analysis of mCD33 expression on primary microglia isolated directly from an adult brain via a Percoll gradient from three independent WT and mCD33−/− mice. **k** Levels of mCD33 expression of microglia derived from the brain of neonatal WT (mCD33+/+; black) and CD33-deficient (mCD33−/−; gray) mice. **l** Levels of mCD33 expression on microglia expanded from the brain of neonatal WT mice cultured with TGF-β1 M-CSF, GM-CSF/ INFγ/LPS, or M-CSF/IL-4/IL-13. Statistical significance calculated based on an unpaired Student's T-test.

Fig. 1f), T cells (CD3+ CD19− CD11b−; Fig. 1g), eosinophils (CD11b+ B220− CD11c− Ly-6Clow; Fig. 1h), and macrophages (F4/80++ Cx₃cr1+ MHCII+ CD11blow CD11clow Ly-6Clow B220−; Fig. 1i). In the brain, low levels of mCD33 were observed on microglia (CD11b+ Cx₃cr1+ F4/80+ Ly-6C− Ly-6G−; Fig. 1j) isolated directly from adult mice. Low levels of mCD33 were also found on microglia derived from the brain of neonatal mice (Fig. 1k). Increased mCD33 cell surface levels were observed upon polarization of these neonatal microglia with GM-CSF followed by INFγ and LPS, but not with TGF-β1 and M-CSF or M-CSF followed by IL-4 and IL-13 (Fig. 1l). Transcript levels of mCD33 measured under these same polarization conditions support

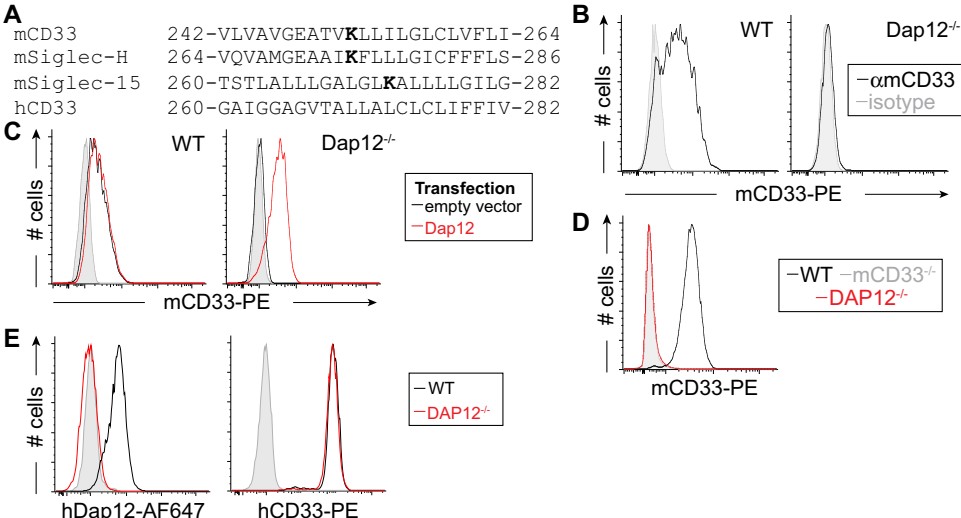

**Fig. 2** Dap12-dependent cell surface expression of mCD33. **a** Sequence alignment of the transmembrane segment of three murine Siglecs bearing a transmembrane lysine residue, shown in bold, along with hCD33. **b** Flow cytometry analysis of mCD33 expression on WT and CRISPR/Cas9-generated Dap12$^{-/-}$ RAW264.7 cells (gray: isotype control; black: αmCD33). **c** WT and Dap12$^{-/-}$ RAW264.7 cells were transiently transfected with mDap12 (red) or empty vector (black) and stained with αmCD33 (black and red) or isotype (gray). **d** Expression of mCD33 on WT (black), mCD33$^{-/-}$ (gray), and Dap12$^{-/-}$ (red) peripheral blood neutrophils (CD11b$^+$Ly-6G$^+$Ly-6C$^-$). **e** Expression analysis of Dap12 and hCD33 on WT (black) and Dap12$^{-/-}$ U937 cells (red); isotype is shown in gray and was at the same levels for both cell types.

upregulation of mCD33 expression under the conditions of GM-CSF followed by INFγ and LPS (Supplementary Fig. 2).

**Dap12-dependent expression of mCD33.** A positively charged residue in the transmembrane region of immune co-receptors often signifies pairing with an adapter protein containing a complementary aspartic acid residue[7]. Such a lysine is found in mCD33 at amino acid 252, which falls within the predicted transmembrane sequence, and is similar to Siglec-H and Siglec-15, but is notably absent in hCD33 (Fig. 2a). Cell surface expression of Siglec-H and Siglec-15 are dependent on Dap12[8,9], therefore, to assess a requirement of Dap12 for mCD33 expression, we used CRISPR/Cas9 to delete Dap12 from the RAW264.7 murine macrophage cell line. We find that expression of mCD33 cells is completely lost upon deletion of Dap12 (Fig. 2b). Transient transfection of Dap12 in Dap12$^{-/-}$ cells restored mCD33 cell surface levels, demonstrating that loss of mCD33 on the cell surface upon deletion of Dap12 was not due to an off-target effect of CRISPR/Cas9 (Fig. 2c). We also assessed mCD33 expression on peripheral blood neutrophils from Dap12$^{-/-}$ mice, which recapitulated the findings that mCD33 cannot be detected on the cell surface in the absence of Dap12 (Fig. 2d). Contrasting with these results in mouse cells, deletion of Dap12 from human U937 monocytic cells had no effect on the expression levels of hCD33 (Fig. 2e). These findings support mCD33 as a Dap12-paired receptor, which is a feature not shared with hCD33.

**mCD33 does not regulate cargo uptake in cultured cells.** To investigate a role for mCD33 in regulating cargo uptake, we used CRISPR/Cas9 to generate mCD33-deficient RAW264.7 (Fig. 3a) and BV-2 cells (Fig. 3b). To minimize effects that could stem from clonal variability in functional studies, we isolated nine mCD33$^{-/-}$ clones and six mCD33$^{+/+}$ clones of RAW264.7 cells, as well as six mCD33$^{-/-}$ and mCD33$^{+/+}$ of BV-2 cell clones. In the RAW264.7 cells, uptake of four different types of fluorescent cargo was monitored by flow cytometry, including TRITC-dextran (Fig. 3c), blue fluorophore-labeled polystyrene beads (Fig. 3d), pHrodo-myelin (Fig. 3e), and HyLight555 aggregated Aβ$_{1-42}$ (Fig. 3f). Loss of mCD33 did not impair the ability of RAW264.7 to take up any

of the four cargo. To ensure that we were monitoring internalization in these studies, we performed a time courses of uptake of pHrodo-myelin at 4 and 37 °C, which revealed time-dependent increases at 37 °C with no conditions where significant differences are observed between mCD33$^{+/+}$ and mCD33$^{-/-}$ cells (Supplementary Fig. 3). We also used another readout of cellular uptake to corroborate these findings by visualizing cargo cellular uptake of the polystyrene beads by microscopy. By masking the cell boundary with a cell-penetrating fluorophore, the % of cells with at least one bead inside the cell was visualized and quantified (Fig. 3g). Using this approach, no difference in % phagocytosis was observed between six mCD33$^{+/+}$ and six mCD33$^{-/-}$ RAW264.7 clones in an individual experiment (Fig. 3h) or over five independent experiments (Fig. 3i). In BV-2 cells, uptake of polystyrene beads (Fig. 3j) and aggregated Aβ$_{1-42}$ (Fig. 3k) also showed no mCD33-dependent effect in flow cytometry-based uptake assays, with Cytochalasin-D blocking a significant amount of the uptake.

**mCD33 does not regulate cargo uptake in mouse microglia.** To compare phagocytosis from mCD33$^{+/+}$ and mCD33$^{-/-}$ microglia isolated directly from adult mice, a competitive phagocytic assay was developed since, on average, only 20–30 K microglia can be isolated from a single mouse brain. Since mCD33 expression levels were insufficient to distinguish the mCD33$^{+/+}$ and mCD33$^{-/-}$ microglia, mCD33$^{+/+}$ and mCD33$^{-/-}$ microglia were tested separately but in competition with age- and sex-matched mCD33$^{+/+}$ microglia from CD45.1$^+$ mice. In this way, CD45.1 vs CD45.2 staining could be used to differentiate the two types of cells (Fig. 4a). Accordingly, the % phagocytosis of the CD45.2$^+$ WT or CD33$^{-/-}$ microglia was set as a percentage of the WT CD45.1$^+$ microglia and overall revealed no difference in uptake of the four different types of cargo for the mCD33$^{+/+}$ and mCD33$^{-/-}$ microglia (Fig. 4b–e). We also carried out parallel phagocytosis assays with microglia expanded from the brain of mCD33$^{+/+}$ and mCD33$^{-/-}$ neonatal mice. No significant difference between mCD33$^{+/+}$ and mCD33$^{-/-}$ microglia was observed with aggregated Aβ$_{1-42}$ (Supplementary Fig. 4). Since polarization conditions that include GM-CSF/INFγ/LPS led to upregulated expression of mCD33 (Fig. 2l), we also performed

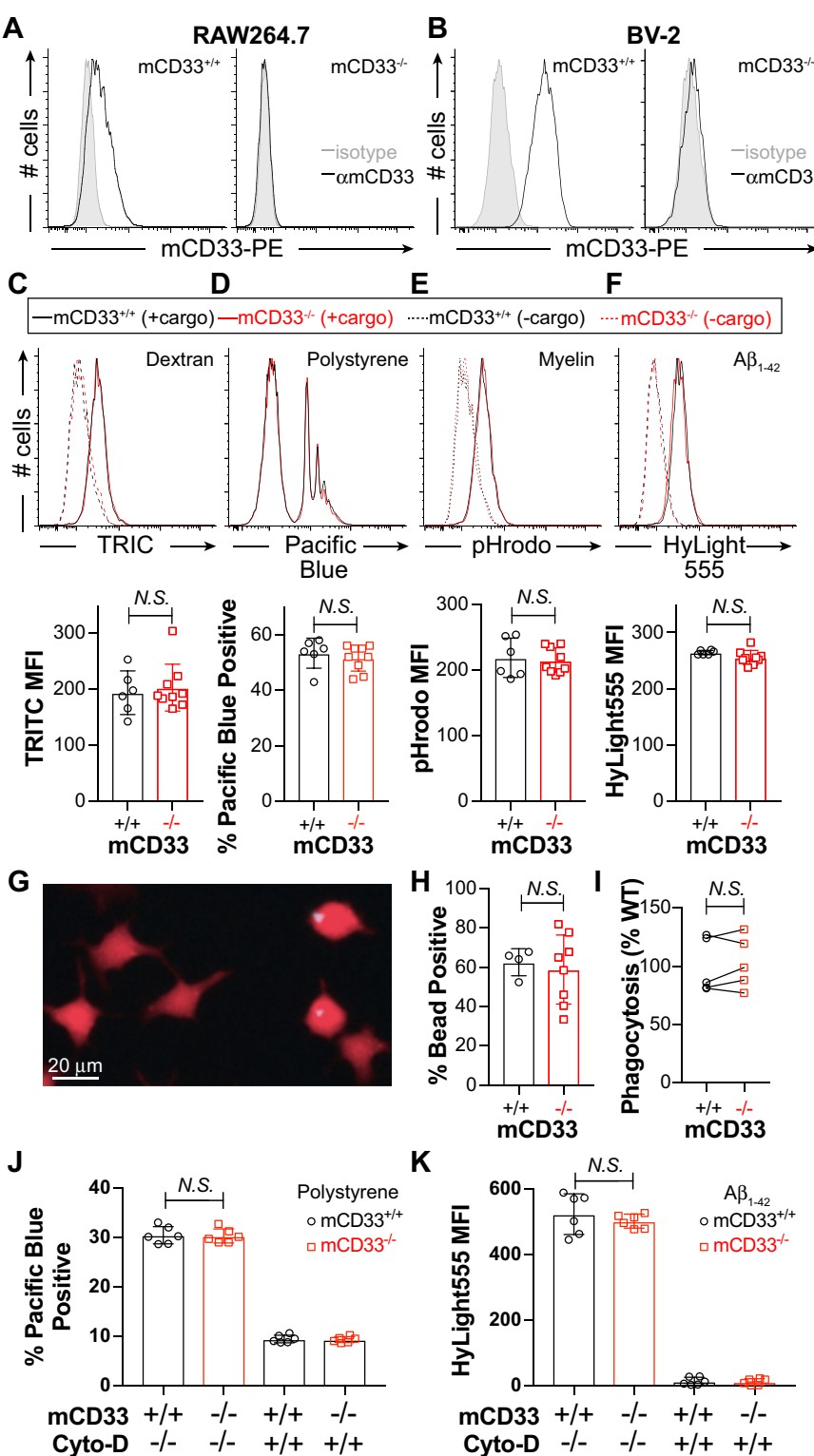

cellular uptake under these differentiated conditions but, again, no mCD33-dependent effects were observed in polystyrene bead or Aβ$_{1-42}$ uptake, both of which was strongly inhibited by Cytochalasin-D (Fig. 4f, g).

**Enhanced uptake of cargo due to loss of hCD33 expression**. THP-1 and U937, which both highly express hCD33M, were used as human phagocytic cell lines. Using CRISPR/Cas9, variants of both cell lines lacking hCD33 expression were isolated (Fig. 5a, b). Specifically, we isolated seven different hCD33$^{-/-}$ and hCD33$^{+/+}$ U937 clones, along with three clones of hCD33$^{-/-}$ and hCD33$^{+/+}$ THP-1 cells. All clones were analyzed in flow cytometry-based uptake assays. In U937 cells, we find that uptake of dextran (Fig. 5c), polystyrene beads (Fig. 5d), myelin (Fig. 5e), and aggregated Aβ$_{1-42}$ (Fig. 5f) were significantly and consistently increased in all seven of the clones of U937 cells with disrupted hCD33 expression. Time- and temperature-dependent uptake of

**Fig. 3 Phagocytosis in mCD33$^{+/+}$ and mCD33$^{-/-}$ cultured macrophages and microglia. a, b** Abrogated expression of mCD33 expression in **a** RAW264.7, **b** BV-2 cells by flow cytometry. **c–f** Flow cytometry-based analysis of **c** dextran particles, **d** polystyrene beads, **e** myelin, **f**, and aggregated Aβ$_{1-42}$ in mCD33$^{+/+}$ (black; six independent clones) and mCD33$^{-/-}$ (red; nine independent clones) RAW264.7 cells. For each cargo, a representative flow cytometry data and summary plots for each clone, where each data point represents the average of at least three replicates for each clone. Note that for the polystyrene beads, cells incubated without beads are not shown as they overlay directly under the major peak on the left, which represent cells that have not taken up the beads. Statistical significance calculated based on an unpaired Student's T-test. **g–i** Microscopy-based analysis of polystyrene bead uptake in mCD33$^{+/+}$ ($n = 4$ independent clones) and mCD33$^{-/-}$ ($n = 8$ independent clones) RAW264.7 clones. **g** Representative image of cells imaged following uptake (red = Calcein, blue = polystyrene bead). **h** Results for a single experiment where results for each clone are average for three different wells. **i** Summary of five independent experiments setting the average levels in the WT clones to 100%; *N.S.* represents no statistical significance based on a paired Student's T-test. **j, k** Phagocytosis of polystyrene beads (**j**) and aggregated Aβ$_{1-42}$ (**k**) in mCD33$^{+/+}$ (black; six independent clones) and mCD33$^{-/-}$ (red; six independent clones) BV-2 cells in the absence and presence of 10 μM Cytochalasin-D. Statistical significance calculated based on an unpaired Student's T-test.

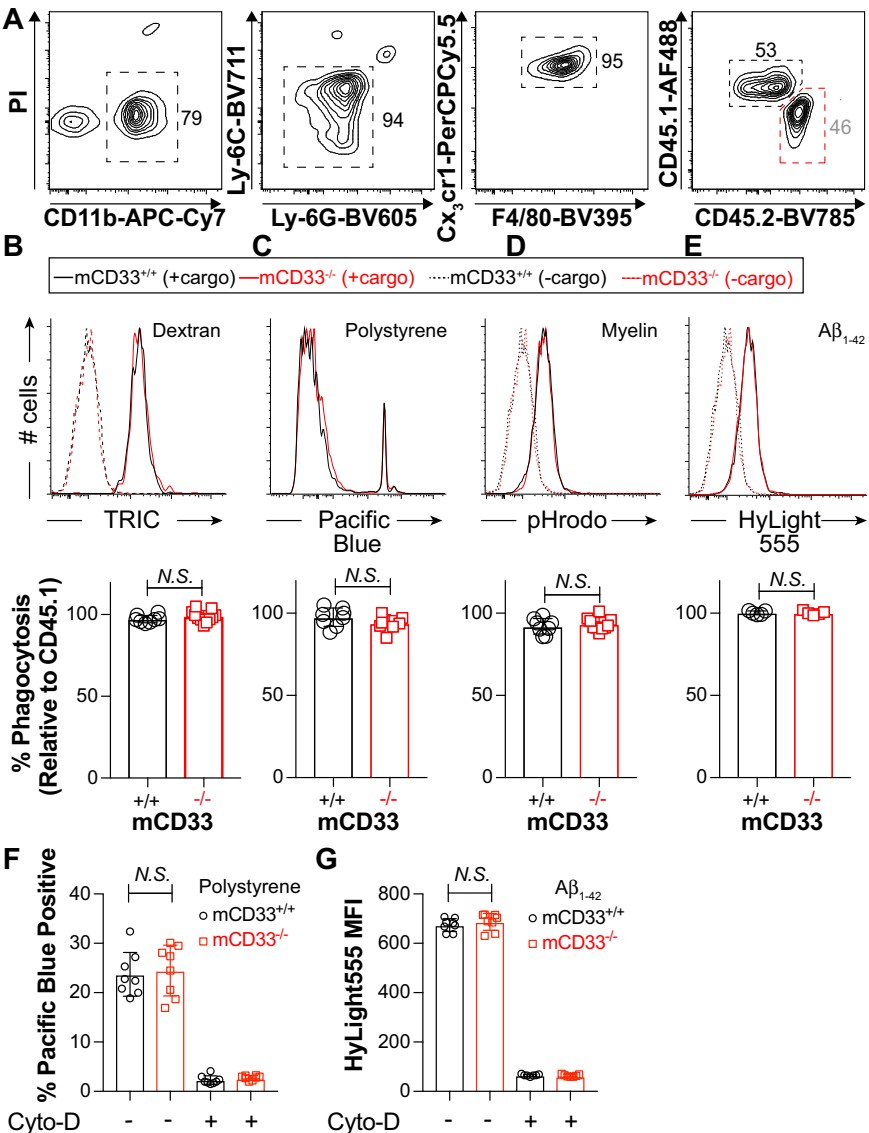

**Fig. 4 A competitive phagocytic assay to examine cargo uptake in WT and mCD33$^{-/-}$ primary microglia. a** Gating strategy for a flow cytometry-based competitive phagocytosis assay in microglia directly isolated from CD45.2$^{+/+}$ WT or mCD33$^{-/-}$ mice tested in competition verses microglia from CD45.1$^{+/+}$ WT mice. **b–e** Results of the competitive flow cytometry-based uptake of **b** dextran particles (11 independent experiments), **c** polystyrene beads (9 independent experiments), **d** myelin (9 independent experiments), and **e** aggregated Aβ$_{1-42}$ (5 independent experiments). Shown are representative flow cytometry histograms for WT (CD45.1$^+$) versus mCD33$^{-/-}$ (CD45.2$^+$) and summary plots for each clone for each genotype plotted as a percentage compared with WT CD45.1$^+$ cells. **f, g** Flow cytometry-based phagocytosis of polystyrene beads (**f**) and aggregated Aβ$_{1-42}$ (**g**) from WT and mCD33$^{-/-}$ microglia expanded from the brain of neonatal mice polarized with GM-CSF/INFγ/LPS where each point represents a different mouse ($n = 8$). *N.S.* represents no statistical significance based on an unpaired Student's T-test. All error bars represent +/− the standard deviation.

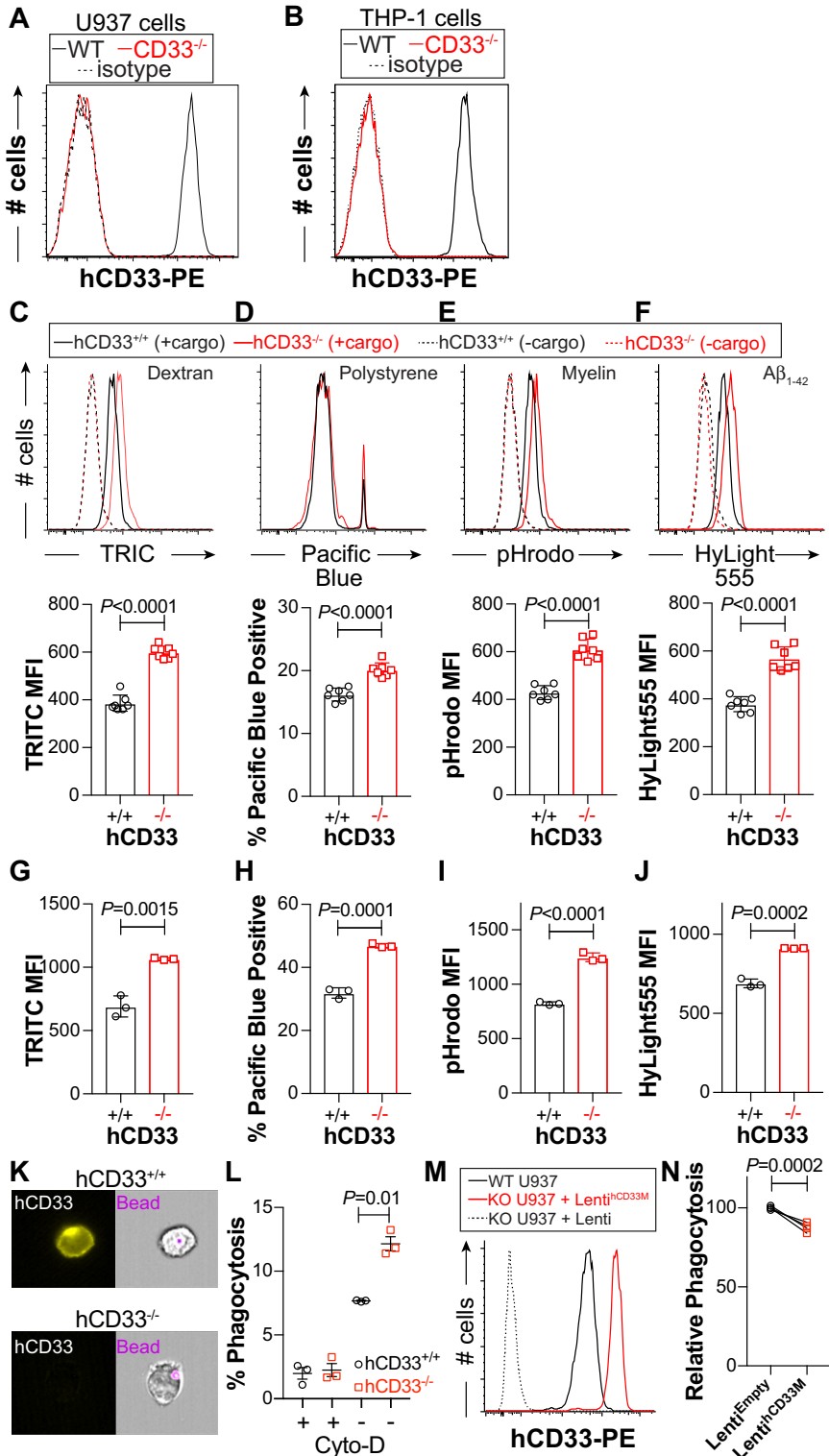

**Fig. 5** hCD33 negatively regulates cargo uptake in human monocytic cell lines. Targeted deletion of CD33 by CRISPR-Cas9 in **a** U937 and **b** THP-1 cells as demonstrated by flow cytometry staining with αhCD33 (clone WM53). Seven different clones of U937 (**c-f**) or three different clones of THP-1 (**g-j**) were tested in flow cytometry-based uptake of **c**, **g** dextran particles, **d**, **h** polystyrene beads, **e**, **i** myelin, and **f**, **j** aggregated Aβ$_{1-42}$. For U937 cells (**c-f**), a representative flow cytometry histogram is shown for each cargo and summary plot of each clone. **k** Representative images of phagocytosis of polystyrene beads by U937 cells visualized by imaging flow cytometry. **l** In three independent experiments, hCD33$^{-/-}$ U937 cells have a higher percentage of cells with internalized polystyrene beads. **m** Lentiviral expression of hCD33M in hCD33$^{-/-}$ cells demonstrating that hCD33 expression levels (red) are higher than those of the parent parental U937 cell line expressing endogenous hCD33 (black). Statistical significance based on an unpaired Student's T-test. **n** Three independent stable cell lines transduced to express empty lentivirus (black) or hCD33M (red) were analysis for phagocytosis with polystyrene beads by flow cytometry. Statistical significance based on a paired Student's T-test.

pHrodo-myelin by U937 cells showed that hCD33$^{-/-}$ cells had increased uptake compared with hCD33$^{+/+}$, which was unique to 37 °C (Supplementary Fig. 5). A similar effect in cargo uptake was observed in hCD33$^{-/-}$ THP-1 clones relative to their hCD33$^{+/+}$ counterparts (Fig. 5g–j). To more directly demonstrate phagocytosis of the polystyrene beads, we used imaging flow cytometry to quantify the % of cells with a fluorescent bead inside of U937 cells (Fig. 5k). The assay was carried out in competition with hCD33$^{-/-}$ and hCD33$^{+/+}$ U937 cells, with results clearly demonstrating increased phagocytosis in the hCD33$^{-/-}$ cells that was ablated by Cytochalasin-D (Fig. 5l).

The gRNA used to disrupt the gene encoding CD33 in U937 and THP-1 cells targets the DNA encoding the D2 domain of hCD33, meaning that expression of both the short (hCD33m) and long (hCD33M) isoforms will be lost. To specifically demonstrate a role for hCD33M in the repression of cargo uptake, we genetically complemented hCD33$^{-/-}$ U937 cells with hCD33M using lentiviral transduction to restore expression of hCD33M (Fig. 5m). In three independent rounds of transduction, hCD33$^{-/-}$ cells transduced to express hCD33M demonstrated a blunted ability to phagocytosis the polystyrene beads compared with cells transduced with control lentivirus (Fig. 5n). The same approach was used to try and overexpress mCD33 in hCD33$^{-/-}$ U937 cells, however, only low levels of mCD33 on the cell surface could be achieved (Supplementary Fig. 5A). Consistent with the amount of WT mCD33 on the cell surface being limited by its Dap12-dependency, a transduction of lentivirus to express a K252A mutant of mCD33 achieved high levels of expression (Supplementary Fig. 6A). Importantly, neither WT nor the highly expressed K252A variant of mCD33 suppressed phagocytosis (Supplementary Fig. 6B). These results in monocytic cell lines demonstrate that loss of hCD33 expression results in a 20–50% increase in cellular uptake of cargo, with genetic complementation of hCD33M restoring phagocytosis.

**Dampened cargo uptake in mouse microglia expressing hCD33M.** Recently we described a mouse model in which the cDNA encoding the long isoform of hCD33 (hCD33M) was inserted into the Rosa26 locus under control of Cre recombinase[38]. This previous study investigated hCD33 in mast cells, but here we took advantage of Cx$_3$cr1$^{cre}$ mice to enable expression of hCD33M in microglia[39]. Competitive phagocytosis assays were carried out between control (Cx$_3$cr1$^{Cre+/-}$ hCD33M$^{-/-}$) and hCD33M transgenic (hCD33M-Tg; Cx$_3$cr1$^{Cre+/-}$ hCD33M$^{+/-}$) microglia by taking advantage of the differential expression of bicistronic expression of GFP and hCD33M (Fig. 6a). Using the four aforementioned fluorescent cargo, we observed a consistent 10–30% decrease in the uptake of the fluorescent cargo in the microglia expressing hCD33M as compared with the WT microglia (Fig. 6b–e). We also assessed microglia expanded from the brain of neonatal mice and found that hCD33M similarly repressed uptake of polystyrene beads and aggregated Aβ$_{1-42}$ (Supplementary Fig. 7). We additionally crossed the hCD33M-Tg mice onto a mCD33$^{-/-}$ background and carried out competitive uptake assays between mCD33$^{-/-}$ (mCD33$^{-/-}$ Cx$_3$cr1$^{Cre+/-}$ hCD33M$^{-/-}$) and mCD33$^{-/-}$ hCD33M-Tg (mCD33$^{-/-}$ Cx$_3$cr1$^{Cre+/-}$ hCD33M$^{+/-}$) microglia, which revealed that expression of hCD33 still resulted in a significant decrease in uptake of all four cargo (Fig. 6f–i). If hCD33 and mCD33 were repressing the same pathways, a larger repressive effect from hCD33 would be anticipated on a mCD33$^{-/-}$ background, however, this was not observed. These results provide further support for hCD33M as a negative regulator of phagocytosis, which is a function not shared by its murine counterpart.

**Discussion**
The strong genetic link between variants of CD33 and AD susceptibility suggests that targeting the common risk allele of CD33, which preferentially encodes the longer isoform (hCD33M) containing its glycan-binding domain, could be a treatment strategy in neurodegenerative disease. To better understand if targeting CD33 in AD is a viable option, a better grasp is needed on the role CD33 plays in modulating the function of microglia. Our findings demonstrate that expression of the long isoform of hCD33 (hCD33M) alone is sufficient to repress phagocytosis in both monocytes and microglia. Our transgenic mice expressing hCD33M will be a valuable tool for future studies addressing the role of hCD33 in modulating plaque accumulation as well as pre-clinical testing of therapeutics aimed at targeting hCD33.

Previous work demonstrated that primary mouse microglial cells lacking mCD33 show enhanced phagocytosis towards Aβ$_{1-42}$, correlating with decreased plaque accumulation in APP/PS1 and 5XFAD mice lacking mCD33[33,34]. These results suggested a conserved function between murine and human CD33 as a negative regulator of phagocytosis. We find that in cultured RAW264.7 macrophages, BV-2 cultured microglia, as well as primary mouse microglia from neonatal and adult mice, no difference was observed in phagocytic ability between WT and mCD33$^{-/-}$ cells. Moreover, overexpression of mCD33 in hCD33$^{-/-}$ U937 cells also did not result in impaired phagocytosis. The inability of mCD33 to repress phagocytosis may be related to the absence of an ITIM since overexpression of a K252A mutant of mCD33 achieved a high level of expression in hCD33$^{-/-}$ U937 cells yet still did not repress phagocytosis.

Given that these results are not in line with the earlier findings, we were careful to reach these conclusions by using competitive assays whenever possible, assaying numerous clones of cultured cells to avoid clonal variability, using age- and sex-matched primary mouse microglia, and testing four different types of fluorescent cargo. It is difficult to reconcile why our results differ from those reported previously in terms of a putative role for mCD33 in repressing phagocytosis but it is worth noting that commercially-available mCD33$^{-/-}$ mice are not on a pure C57BL/6 background, as evidenced by the appearance of a gene regulating pigmentation (p allele) that lies close to the Cd33 locus, derived from 129 founder mice[15]. We observed white/gray mice at a significant frequently following initial established of this line from the commercial supplier, but after extensive backcrossing (>10 generations) onto pure C57BL/6J mice, this pigmentation issue was eliminated and these backcrossed mice were used in our experiments. It also remains possible that the decreased plaque deposition in mCD33$^{-/-}$ mice is the result of a yet undiscovered role for mCD33 in regulating microglial cell function.

Through development of the first, to our knowledge, reported monoclonal antibody targeting mCD33, we demonstrate that primary microglia do express low levels of mCD33, while in the spleen mCD33 is expressed highest on neutrophils. This latter observation is in line with previous work concluding that granulocytes express the highest levels of mCD33[15], which is also consistent with neutrophils having the highest mCD33 mRNA transcript levels[40]. We also demonstrated, for the first time to our knowledge, that cell surface expression of mCD33 on RAW264.7 cells and primary mouse neutrophils is dependent on Dap12 expression, which strongly suggests pairing of the two proteins. This phenomenon was not observed for hCD33. Thus, in mice CD33, Siglec-H, and Siglec-15 are the Dap12 associated Siglecs[8,9]. A functional role for mCD33 in regulating immune cells still remains elusive, but demonstrating that mCD33 pairs with Dap12 opens the door for further research, including a potential activatory role for mCD33.

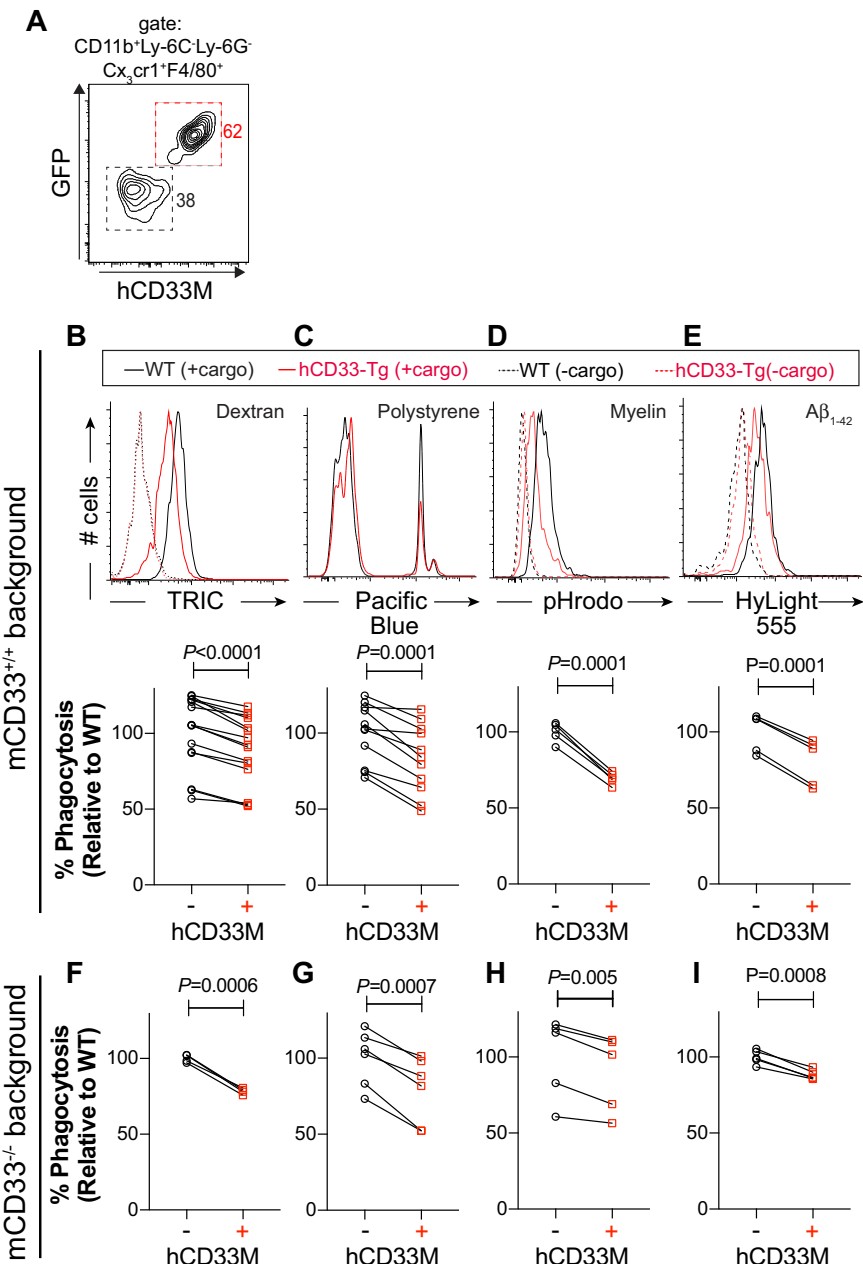

**Fig. 6** Transgenic expression of hCD33M in mouse microglial dampens phagocytosis. **a** Flow cytometry-based competitive phagocytosis gating strategy for WT (GFP⁻hCD33⁻) and hCD33M-Tg (GFP+hCD33M⁺) microglia. The competitive phagocytotic assay was carried out with hCD33M-Tg microglia on a mCD33⁺/⁺ background (**b–e**) or mCD33⁻/⁻ background (**f–i**) where the non-transgenic (hCD33M⁻) microglia are WT or mCD33⁻/⁻, respectively. Flow cytometry-based uptake of **b**, **f** dextran particles (14 independent experiments for mCD33⁺/⁺ and 4 independent experiments for mCD33⁻/⁻), **c**, **g** polystyrene beads (11 independent experiments and 6 independent experiments for mCD33⁻/⁻), **d**, **h** myelin (5 independent experiments and 5 independent experiments for mCD33⁻/⁻), and **e**, **i** aggregated Aβ₁₋₄₂ (5 independent experiments and 5 independent experiments for mCD33⁻/⁻). For cells on a mCD33⁺/⁺ background (**b–e**), a representative flow cytometry histogram is shown for each cargo and summary plots for each assay, where the average value for the hCD33 non-expressing cells was set to 100%. Statistical significance based on a paired Student's T-test.

Decreased expression levels of hCD33M in monocytes bearing the minor protective *CD33* allele correlates with enhanced phagocytosis[21,26,33]. However, it cannot be ruled out that this phenotype is influenced by increased expression of the shorter isoform of CD33 (hCD33m) generated from exon 2 skipping. We found that loss of hCD33 expression in two monocytic cell lines led to consistent increases in uptake of fluorescent cargo, while transgenic expression of hCD33M in mouse microglia or genetic complementation of hCD33M in hCD33⁻/⁻ U937 cells gave rise to suppression of cargo uptake. These findings strongly support a

role for hCD33M in repressing uptake of cargo within phagocytic cells. Future studies are needed to determine what role hCD33m has in phagocytosis.

In our studies, all four types of fluorescence cargo showed similar hCD33-dependent effects in repressing their uptake, despite having significantly different properties, the most notable being size. The dextran and aggregated Aβ used in our studies have reported radii of 4.5 and 3 nm, respectively[41,42]. Particles of this size are typically thought to enter cells via a mechanism involving clathrin-dependent pinocytosis[43]. On the other hand,

the myelin and polystyrene beads used in our studies are much larger, with radii 1 μm or larger[44]. Particles of this size are taken up by phagocytosis[45]. The ability of hCD33M to regulate uptake of all four cargo may have important implications for its mechanism of action, which clearly warrants further study given that this role could be related to AD susceptibility.

Accumulation of aggregated $A\beta_{1-42}$ drives the formation of amyloid plaques and mouse models to study human genetic factors that modulate this process in vivo have been widely used[46]. Our studies suggest that mCD33 may not be an appropriate surrogate for studying hCD33. We demonstrate that transgenic expression of hCD33M in the microglial cell lineage inhibits phagocytosis; these hCD33M transgenic mice should provide a valuable model to test the role of hCD33M in regulating plaque accumulation in vivo, which is currently being tested in ongoing studies in our laboratory. Indeed, establishing a good mouse model to study hCD33 is critical for both better understanding AD pathology and also testing therapeutics aimed at controlling microglial cell function by targeting hCD33M.

## Methods

**Mice and animal care**. All mice were on a C57BL/6J genetic background. WT, CD45.1[+/+], $Cx_3cr1^{Cre}$ [39], and mCD33[−/−] [15] mice were obtained from The Jackson Laboratory. Transgenic mice expressing hCD33M in the Rosa26 locus (R26-hCD33M) were described previously[38]. All animals used were maintained in an access-controlled barrier facility under specific-pathogen-free conditions. Studies were performed in accordance with Public Health Service guidelines and approved by the Animal Subjects Committee of the University of Alberta and the IACUC of The Scripps Research Institute.

**Statistics and reproducibility**. A Student's t-test was used to access statistical significance. When experiments were carried out in competition, a paired Student's t-test was used, while an unpaired Student's t-test was used in cases where experiments groups were analyzed independently. For experiments using cultured cells to assess phagocytosis, three independent replicates of the experiments were performed with representative results from one of the replicates shown. In some instances, a summary of independent experiments is reported to demonstrate the level of experimental variability and reproducibility of the findings.

**Development of a monoclonal antibody targeting mCD33**. We anticipated that the transmembrane lysine residue in mCD33 could be an issue for cell surface expression, we used a strategy previously employed to express Siglec-15 on the surface CHO cells[47], which involves swapping out the transmembrane and cytoplasmic tail of mCD33 with that from mCD22, leaving the extracellular portion of mCD33 (residues 1–240) unchanged. This construct was expressed in CHO and HEK293 cells as the immunogen and bait for screening supernatant from hybridomas, respectively. Accordingly, CHO cells expressing this mCD33 construct were immunized in Rats in the same was as described previously to generate an αSiglec-F monoclonal antibody[37]. Isotyping of the antibody took place through incubation of supernatant of the αmCD33-expressing hybridoma with CHO cells expressing the mCD33 construct, followed by staining with different PE-conjugated secondary antibodies and analysis by flow cytometry. This mCD33 antibody (clone 9A11; originally licensed to eBiosciences, now Thermo Fisher) was used in assessing mCD33 expression on immune cell subsets in its PE-conjugated form.

**Cell lines**. THP-1, U937, RAW264.7, and BV-2 cells were obtained from ATCC. RAW264.7 and BV-2 cells were cultured in DMEM containing 10% fetal bovine serum (FBS; Gibco), 100 U/ml Penicillin, and 100 μg/ml Streptomycin (Gibco). RAW264.7 cells were carefully maintained at a passage number <30 since going beyond this leads to variability in phagocytosis[48]. U937 and THP-1 cells were grown in RPMI (Life Technologies) containing 10% FBS, 100 U/ml Penicillin and 100 μg/ml Streptomycin.

**Flow cytometry and microscopy**. All flow cytometry data were collected on a 5-laser Fortessa X-20 (BD Bioscience) and analyzed using FlowJo software (BD Biosciences). Images to quantitate phagocytosis in RAW cells and brain sections were acquired on an ImageXpressPico (Molecular Devices) at ×20 and ×40 magnification, respectively, and analysis using CellReporterXpress (Molecular Devices). Cell sorting took place on an Aria III (BD Bioscience). Imaging flow cytometry was carried out on an AMNIS ImageStreamx Mark-II Flow Cytometry system and analyzed using Inspire[TM] software.

**Isolation of adult mouse microglia**. Adult mice were euthanized under $CO_2$. Media used for the primary microglia consisted RPMI with 10% FBS, 100 U/mL

Penicillin and 100 μg/ml Streptomycin. Isolated brain samples were homogenized by a 5 ml syringe plungers in media through 40 μm corning filter units under sterile condition. Homogenized samples were centrifuged at 500 rcf for 5 min and the pellet was treated with 3 ml of red blood cell lysis buffer (150 mM $NH_4Cl$, 9 mM $NaHCO_3$, and 0.1 mM EDTA). Following centrifugation at 300 rcf for 5 min, the pellet was dissolved in 3 ml of 30% Percoll (Percoll PLUS, GE Healthcare) and carefully layered on top of 70% Percoll and immediately centrifuged (650 rcf for 20 min). Immune cells were isolated from the border between the two layers, washed (300 rcf, 5 min) and resuspended in media.

**Analysis of mCD33 expression on primary immune cells**. mCD33 expression was accessed in WT and mCD33[−/−] mice from within the spleen and brain. For the spleen, it was homogenized in media through a 40 μm filter (Corning) under sterile conditions. Samples were centrifuged at 300 rcf for 5 min and the pellet was treated with 3 ml RBC lysis buffer (1 min). Splenocytes or immune cells isolated from the brain (see above) were centrifuged at 300 rcf for 5 min and treated with 25 μl mouse Fc-Block (1:50 dilution) antibody (TruStain fcx[TM] anti-mouse clone 93, BioLegend) in flow buffer (HBSS containing 0.1% BSA and 2 mM EDTA). Fc-Block antibody treated samples finally stained with 25 μl antibody cocktail. For analysis of splenocytes, the following cocktail was used: markers of CD11c (AF488, clone N418, BioLegend), CD19 (BV785, clone 6D5, BioLegend), CD3 (BV650, clone 17A2, BioLegend), B220 (BUV395, clone 6B2, BD Horizon), NK1.1 (BV421, clone PK136, BioLegend), mCD33 (PE, clone 9A11, eBioscience), CX3CR1(AlexaFluor 647, clone SA011F11, BioLegend), Ly-6C (APC/Cy7, clone HK 1.4, BioLegend), Ly-6G (PerCP/Cy5.5, clone 1A8, BioLegend), F4/80 (BV711, clone T45-2342, BD Horizon), Cd11b (BV510, clone M1/70, BioLegend), MHCII (BV605, clone M5/114,15.2, BioLegend), and Siglec-H (PE-Cy7, clone eBio440C, Invitrogen). For analysis of microglia, the following cocktail was used: F4/80 (BUV395, clone T45-2342, BD Horizon), LY-6G (BV605, clone 1A8, BioLegend), CD11b (APC/Cy7, clone M1/70, BioLegend), $Cx_3cr1$ (PerCP/Cy5.5, clone SA011F11, BioLegend), and Ly-6C (BV711, clone HK 1.4, BioLegend). Cells were stained at 4 °C for 20 min, washed once, and resuspended in flow buffer containing 1 μg/ml propidium iodide prior to analysis by flow cytometry.

**CRISPR/Cas9 gene editing**. Custom crRNA (Integrated DNA Technologies; IDT) was designed to target human CD33 (target sequence = GAACACCCCCGATCTTCTCC), mouse CD33 (target sequence = AGGTGTGAACGTCAGCACGG), mouse Dap12 (target sequence = GGGGCTGGAGGGGGCTGGTC), and human Dap12 (target sequence = TGAGACCGAGTCGCCTTATC). Cells were seeded at 500,000 cells per well 1 day prior to transfection in a 6-well tissue culture plate. For one well of a 6-well plate, 6.25 μg of Cas9 nuclease (IDT), 1.2 μg of ATTO-550 labeled crRNA:tracrRNA (IDT) duplex, 12.5 μg Cas9 Plus reagent (IDT), and 7.5 μL CRISPRMAX reagent (Thermo Fisher) in 250 μL Opti-MEM medium (Gibco). One day after transfection, cells were removed from the plate—in the case of RAW cells—by trypsin digestion, washed, resuspended in 300 μl of cell sorting medium (HBSS, 10% FBS, 1 mM EDTA), and stored on ice until sorting. Cells were sorted within the University of Alberta Flow Cytometry Core. The top 5% bright dyes stained with ATTO-550 were sorted into three 96-well plates containing regular culture medium at one or three cells per well for U937 and RAW cell lines, respectively. Cells were grown for ~3 weeks until a time when colonies were larger enough to be transferred to a 12-well plate. Once cells were 90% confluent cells, they were screened by flow cytometry using PE-conjugated antibody for either hCD33 (clone WM53) or mCD33 (clone 9A11). For knockout of mouse Dap12, RAW264.7 clones were initially screen for mCD33 expression and one clone that stained negative for mCD33 was taken for sequence validation, which involved PCR amplification of a 575 bp fragment surrounding the targeted sequence followed Sanger sequencing that validated insertion/deletion at the correct location. For knockout of human Dap12, U937 cells were screened for Dap12 expression by intracellular flow cytometry staining with PE-labeled anti-Dap12 (clone REA900; Miltenyi).

**Transfection of cells with mDap12**. The cDNA encoding mDap12 was commercially synthesized (IDT) and obtained in pcDNA3. RAW264.7 cells were seeded in 6-well dishes the day before transfection at $0.5 \times 10^6$ cells/well. Cells were transfected with 2 μg of DNA using Lipofectamine LTX, according to the manufacturer's protocol (Thermo Fisher). The following day, fresh media was added to the transfected cells and 2 days post transfection cells were removed from the plate with 5 mM EDTA and analyzed for mCD33 expression by flow cytometry.

**Materials for phagocytosis assays**. The follow materials were purchased from commercial suppliers: TRITC-dextran (Tetramethylrhodamine isocyanate-dextran, Sigma, USA), Polystyrene beads (Fluospheres carboxylate-modified 1.0 μm blue 350/440, Molecular Probes, Life Technologies), and Amyloid-beta peptide ($A\beta_{1-42}$, HiLyte Fluor 555-labeled, Anaspec Peptide, USA). Aggregation of Aβ was carried out similar to a published protocol[49]. Fluorescently labeled $A\beta_{1-42}$ (1 mg) was dissolved in Hexafluoro-2-propanol (HFIP) at 1 mM to break up existing aggregates. The HFIP was evaporated in a fume hood overnight, dissolved at 400 μM in DMSO, and stored in 5 μl aliquots at −80 °C. Frozen aliquots were dissolved in HEPES buffered saline (Sigma) to 20 μM and aggregated at 37 °C for 2 days in the

dark, aliquoted and stored at −80 °C. On the day of the experiment, an aliquot was thawed, diluted to 400 nM into cell culture media, and sonicated for 5 min in a water bath sonicator prior to use in the phagocytosis assay.

**Myelin isolation, purification, characterization, and pHrodo labeling**. The brain of a 5-month Sprague Dawley rat was isolated, under anesthesia, and immediately stored at −80 °C in an airtight container. Brains were weighed and processed for myelin isolation according a previous work[50]. Briefly, Brain was minced and homogenized in ice cold 0.3 M sucrose solution. The homogenate was layered over 0.83 M sucrose solution and ultracentrifuged (75,000 rcf, 45 min, 4 °C). The layers of crude myelin, which was formed at the interface of the two sucrose solutions, were collected and subjected to osmotic shock by resuspension in Tris·Cl buffer. The crude myelin was homogenized and was ultracentrifuged two more times (12,000 rcf, 15 and 10 min, 4 °C). Myelin was further purified from the crude preparation by another density gradient and subjected to osmotic shock as above. The protein and lipid contents of myelin was characterized by Western blotting and HPLC. MBP (Myelin basic protein) was detected in the sample in high amounts, while there was no evidence of degradation of lipids or proteins. Myelin was labeled with pHrodo (Invitrogen) by resuspension in sodium bicarbonate 100 mM, pH 8.5 at 100 mg/ml concentration and incubation with 10 μM of dye for 1 h at room temperature. The excess dye was washed out by washing myelin in PBS four times. The labeled myelin was stored in PBS containing 1% DMSO at −80 °C.

**Microscopy-based phagocytosis assay in RAW264.7 cells**. RAW264.7 cell clones ($n = 4$ mCD33$^{+/+}$; $n = 8$ mCD33$^{-/-}$) were grown to ~80% confluency and removed with 5 mM EDTA in PBS. Cell numbers were standardized by cell counting using a hemocytometer. Approximately 25,000 cells were plated, in triplicate, in 96-well flat bottom plate and allowed to adhere and grow in 300 μl of media for 24 h. The following day, media was gently aspirated and 200 μl of media containing 5 μg/ml Calcein AM (Thermo Fisher) and carboxylate fluorosphere beads (1:625–1:2500 dilution) were added. Plates were incubated at 37 °C incubator for 30 min, washed with media three times, followed by image capture. To analyze the data, a cell mask was created for Calcein and software (Molecular Devices) was used to calculate cells that were either positive or negative the presence of carboxylate fluorospheres. Between 2000 and 10,000 cells were analyzed per well. The mean of the triplicates for each clone was taken and plotted as an individual point. Data were normalized according the mean of the WT clones for each experiment.

**Flow cytometry phagocytosis assays in cell lines**. THP-1 and U937 cells were grown to a density of ~$1 \times 10^6$ cells/mL in a T75 flask prior to the assay, harvested, centrifuged, resuspended in media, and 100,000 cells were added to a 96-well U-bottom plates in 100 μl of media. For RAW264.7 and BV-2 cells, cells were grown to 80% confluence, removed from the plate with 5 mM EDTA in PBS, and plated in the same way as the THP-1 and U937 cells. To initiate the assay, 100 μl of media containing the fluorescent cargo was added to the cells and incubated for 30 min at 37 °C. During the incubation step, the following concentration of fluorescent cargos were used: TRITC-dextran, 0.5 mg/ml; polystyrene beads, 1:340 dilution from a 2% stock solution from vendor; pHrodo-myelin (2 mg/ml), and Aβ$_{1-42}$, 400 nM. Following this incubation, cells were washed two times with media, resuspended in flow buffer containing 1 μg/ml propidium iodide, and analyzed immediately by flow cytometry. For the dextran, myelin, and Aβ$_{1-42}$, the extent of phagocytosis was determined by assessing the median fluorescence intensity (MFI) of the fluorescent signal with a background subtraction (cells without cargo), whereas for the polystyrene beads, the % of cells taking up at least one bead was used. To inhibit phagocytosis, cells were pre-treated with 10 μM Cytochalasin-D for 30 min.

**Competitive phagocytosis in primary microglial cells**. To assess microglial phagocytosis in the most control manner, microglia were used from age- and sex-matched adult mice in competitive phagocytic assay. To compare mCD33$^{+/+}$ and mCD33$^{-/-}$ microglia, we could not compare head to head since the levels of mCD33 expression was not sufficient to differentiate. Thus, mCD33$^{+/+}$ and mCD33$^{-/-}$ microglia were separately tested head-to-head with mCD33$^{+/+}$ from CD45.2$^+$ mice. To compare microglia with and without hCD33 expression, cells could be directly compared head-to-head given that the combination of GFP and hCD33 expression easily enabled differentiation. Accordingly, after isolating microglia (Percoll gradient; see above), the appropriate cells were mixed, platted in a V-bottom 96-well plate with 100 μl of media, 100 μl of media containing the appropriate fluorescent cargo was added at the same concentration as described above for cultured cells, and cells were incubated for 30 min at 37 °C. Following a wash, cells were stained with a cocktail of antibodies for 20 min at 4 °C that included: CD11b (APC/Cy7, clone M1/70, BioLegend), Ly-6G (BV605, clone 1A8, BioLegend), Ly-6C (BV711, clone HK 1.4, BioLegend), Cx$_3$cr1 (PerCP/Cy5.5, clone SA011F11, BioLegend), and F4/80 (BUV395 clone T45-2342, BD Horizon). For mCD33$^{-/-}$ versus mCD33$^{+/+}$ analysis, we additionally used the following antibodies: CD45.1 (AF488, clone A20, BioLegend) and CD45.2 (BV785, clone 104, BioLegend). For analysis of microglia expression hCD33, we additionally used the following antibody: hCD33 (PE, clone WM53, BioLegend). Following one wash, cells were resuspended in flow buffer and analyzed by flow cytometry.

**Imaging flow cytometry phagocytosis assay**. hCD33$^{+/+}$ and hCD33$^{-/-}$ U937 cell lines were cultured in complete cell culture medium for at least 48 h prior to the assay. Prior to the assay, the equal numbers of both cells were mixed and incubated with the polystyrene beads for 30 min with or without 10 μM Cytochalasin-D at 37 °C. After the incubation step, cells were washed, stained with PE-labeled anti-hCD33 and AF647-labeled CD11b, washed, and resuspended in 50 μL flow buffer. At least 3000 cells were analyzed. Cell images were acquired at ×60 magnification. To analyze phagocytosis, acquired data were processed using the internalization wizard on the IDEAS software version 6.2. Bright-field images were used to define a mask for the whole cell wherein an internal mask was defined by eroding the whole cell mask by 4 pixels. Doing so enabled the calculation of an internalization score (IS) where a 0.3 IS was set as a threshold for internalization.

**Lentiviral vector production and transduction of U937 cells**. All lentiviral vectors were made using the previously described Lentiviral backbone, RP172[51]. Cloning hCD33M, mCD33, or mCD33$^{K252A}$ into the MCS was done using the restriction sites SphI and PacI. Ligated vectors were transformed into Stable competent *E. coli* (New England Biolabs) to reduce plasmid instability due to recombination. Individual clones were picked and grown up overnight in 7 ml LB media containing 100 μg/ml ampicillin. Following mini preps (Qiagen), Sanger sequencing confirmed the DNA sequence was correct. Production of lentiviral particles was achieved using triple transfection of HEK293T cells with the designed transfer vectors (RP172-hCD33, RP172-mCD33, and RP172-mCD33$^{K252A}$), packaging vector (RP18) containing the viral genes Gag, Pol, and Rev, and an envelope plasmid (RP19) containing VSV-G. Briefly, 150,000 HEK293T cells were plated in 445 μl DMEM growth media (Gibco) containing 10% FBS and Pen/Strep, in a 24-well plate on day 0. On day 1, a transfection mix was made containing 150 ng RP18 (1 μl), 150 ng RP19 (1 μl), 300 ng of transfer vector (2 μl), 1.8 μl TransIT®-LT1 Reagent (Mirus Bio), and 60.2 μl Opti-MEM® media (Gibco). The mix was vortexed and allowed to sit undisturbed for 15 min. This transfection mix (55 μl) was added to wells containing HEK293T cells. The plate was incubated in a tissue culture incubator at 37 °C and 5% CO$_2$. Seventy-two hours post transfection (day 4), the supernatant was harvested, spun down to remove residual cell debris, aliquoted, and stored at −80 °C. U937 cells (150,000) were plated in triplicate in a 24-well plate in 250 μl RPMI growth media (Gibco), containing 10% FBS and Pen/Strep, and 8 μg Polybrene Transfection Reagent (EMD Millipore). A range of virus (1, 10, or 100 μl) was added to each well and incubated for 6–8 h in a tissue culture incubator. After 8 h, the media in each well was topped up to 500 μl. Three days post transduction, cells were harvested and resuspended in flow cytometry buffer (HBSS containing 0.1% BSA and 2 mM EDTA). The titer of each virus was determined by measuring the % of mAmetrine$^+$ cells in each well, in the BV510 channel, by flow cytometry. The dilution factor was multiplied by the starting cell number to calculate a titer, which varied between $1.3 \times 10^7$ and $4.9 \times 10^7$ TU/ml (transduction units). U937 cells (500,000) were plated in a 6-well plate, in 1.3 ml growth media, and 10 μg Polybrene. lentivirus was added to each well at a MOI = 1 (multiplicity of infection), to minimize multiple insertion events in positively transduced cells. Plates were incubated for 24 h then media was topped up to 5 ml. Three days post transduction, cells were wash and resuspended in PBS containing 1 mM EDTA and 1% FBS. Cells were sorted on an Aria cell sorter (BD Biosciences) and mAmetrine$^+$ cells were collected. The number of mAmetrine$^+$ cells in each sample ranged from 0.3 to 1.9%. Sorted cells were plated in 12-well plates and allowed to grow up. As an added measure to ensure success transduction, 300 μg/ml zeocin was added to the cells for 1 week select for transduced cells. Two weeks after the cell sort, >99.5% of cells were mAmetrine$^+$.

**Primary microglia cultures and mRNA analysis**. Primary mixed glial cultures from P0.5-P1.5 mice were prepared following a previously described method[52]. Briefly, cerebral cortices were enzymatically and mechanically dissociated and cortical cells were seeded in DMEM/F12 10% FBS (supplemented with 100 units/ml Penicillin, 100 μg/ml Streptomycin, 1 mM Sodium Pyruvate, and 50 μM β-mercaptoethanol) and cultured at 37 °C, 5% CO$_2$. Growth medium was replaced every 4 days. At day 8, medium was supplemented with GM-CSF 10 ng/ml. At day 14, microglia were harvested by incubating in DMEM/F12 10% FBS 15 mM lidocaine and shaking for 15 min at 200 rpm and further cultured on poly-L-lysine coated plates in DMEM/F12 (supplemented with 1 mM Sodium Pyruvate and 50 μM β-mercaptoethanol)[53]. Cells were let to rest for 24 h before been dissociated from the well using StemPro Accutase (ThermoFisher) for 20 min and collected in HBSS. To examine CD33 under different microglial polarization states, conditions describe previously were used[54]. Specifically, primary microglia were treated during 6 days with TGF-β1 50 ng/ml and M-CSF 10 ng/ml in DMEM/F12 5% FBS; 4 days with GM-CSF, followed by 1 hour with INFγ 20 ng/ml and 2 days with LPS 100 ng/ml; or 4 days with M-CSF 25 ng/ml followed by 2 days with IL-4 20 ng/ml and IL-13 20 ng/ml. To assess CD33 expression by flow cytometry, medium was removed, cells incubated with Accutase for 20 min and collected in HBSS prior to staining with antibodies and flow cytometric analysis. For qPCR analysis of mRNA transcript levels, primary murine microglia, cultured and polarized as described above, were lysed in RLT buffer (QIAGEN). RNA was isolated using RNEasy Micro Kit (QIAGEN) and cDNA was synthetized from ~500 μg of RNA using Oligo dT primers and SuperScript II (ThermoFischer). qPCR was carried out using PowerUp SYBR Green Master Mix (Applied Biosystems) in a StepOne Plus instrument

(Applied Biosystems) and mRNA expression of mRNA was normalized over those of cyclophilin.

**Reporting summary**. Further information on research design is available in the Nature Research Reporting Summary linked to this article.

## Data availability

The authors declare that all data supporting the findings of this study are available within the paper and its supplementary information files. Please contact the corresponding authors (M.S.M.) for access of raw data, which is stored electronically, and will be made available upon reasonable requests.

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

## Acknowledgements

We thank GlycoNet and Eisai for funding through a strategic partnership agreement between Eisai and the University of Alberta. We also thank CIHR for a project grant to M.S. M., the Canadian Foundation for Innovation (CFI) for an award to MM for equipment used in these studies, and the Canada Research Chairs (CRC) program for a Tier II CRC to MM. J.C.P. acknowledges funding from the National Institutes of Health (NIH; HL107151 and AI132790). We thank Lewis Lanier for access to Dap12$^{-/-}$ mice, Susmita Sarker for help with mouse genotyping, and Basil Hubbard for advice and help with CRISPR/Cas9 work. Flow cytometry was performed at the University of Alberta, Faculty of Medicine and Dentistry Flow Cytometry Facility, which receives financial support from the Faculty of Medicine and Dentistry and CFI awards to contributing investigators.

## Author contributions

A.B. and M.S.M. conceived of the project and wrote the paper. E.R. performed all the genome editing. J.J., M.L.-S., J.E., C.S.-L., A.B., and L.S. carried out supporting experiments. S.D., N.K., and J.P. were involved at early stages of the project in raising the antibody and establishing the hCD33M transgenic mouse. M.N.-N. and F.G. provided key reagents and advise on experiments with myelin. F.F. and C.R. provided assistance with getting the lentivirus transduction of U937 cells working. D.G., Q.W., and S.S. prepared the neonatal primary microglia.

## Competing interests

The authors declare no competing interests.
