## [Peer Review File · Communications Biology]

Reviewers' comments:

Reviewer #1 (Remarks to the Author):

Bhattercherjee et al. showed expression of murine CD33 (mCD33) on neutrophils and microglia. Furthermore, Bhattercherjee et al. demonstrated that mCD33 surface expression is dependent on Dap12. Bhattercherjee et al. also observed that deletion of human CD33 (hCD33), but not mCD33, increased phagocytosis in several in vitro models and that transgenic hCD33 expression in mice resulted in a decrease of phagocytosis.

The manuscript is well-written and discussed. Data are of high relevance for understanding the differences between murine and human CD33 signaling.

However, there are some points that should be addressed more carefully:

- The histograms with multiple overlays (Fig. 3-6) are hard to distinguish. The use of colors and/or filled histograms would make it easier to distinguish. Further, in Fig. 6 (B-E and F-I) the labeling related to transgenic hCD33M expression in mice (WT or mCD33^{-/-} background) is missing.
- The subtitles in the results part indicating increased phagocytosis are partly misleading since small size particles such as dextran are taken up via fluid-phase endocytosis or clathrin-dependent endocytosis and thus should not be count to phagocytosis. Moreover, Bhattercherjee et al. did not exclude the possibility of dextran and A β to simple bind to the cells without getting internalized.
- The competitive phagocytosis assay in adult primary CD45.2 WT and mCD33^{-/-} versus CD45.1 is unclear. If mCD33 expression in CD45.2 mice is so little that a knockout does not give reliable results in phagocytosis assays, the choice of the mouse line/cells or the readout is questionable. A more sensitive readout system would be better to detect small changes.
- The statistics of the relative phagocytosis (Fig. 3 H, Fig. 6) appears to be incorrect as the control is always set to 100 without deviation from its mean due to normalization. This increases type I errors in statistics. It would be better to take either raw data or to normalize each value of both, control and experimental group, to the mean of the control group.

Reviewer #2 (Remarks to the Author):

This study by Bhattercherjee et al describes the role human CD33 in phagocytosis, with an emphasis on microglia and abeta clearance in relation to Alzheimer's disease (AD). A major claim is that mice expressing human CD33 in the microglial lineage provide a new and valuable model for AD research. However, the authors do not show this. The least that would be necessary, and very interesting indeed given the fact that abeta in mice does not spontaneously aggregate, is to show that hCD33M expression in microglia in APP/PS1 mice reduces abeta clearance and accelerates plaque deposition. It is crucial to demonstrate this in order to substantiate the claim, in particular also because contradicting findings regarding the role of mCD33 have been published, and all data described are derived from in vitro cellular assays using a single technique.

I also have some concerns regarding the data. It may be my ignorance in the field of flow cytometry, but in all panels where polystyrene phagocytosis is measured, I don't see this back in the flow cytometry charts, which look very different from the other three cargos. This should be explained in

more detail. Also, I don't see solid grey (or blue?) lines in any of the charts, whereas the legends indicate that there should be.

Reviewer #3 (Remarks to the Author):

Brief summary of the manuscript

The authors were interested in understanding whether the CD33 mouse and human receptor orthologs which differ at two positions in the protein, have the same impact on uptake of different cargoes, including A β species. This is of interest because a DNA variant in CD33 is associated with AD risk and establishing the best CD33 AD mouse model able to capture the functional pathway of CD33 and relevant to Alzheimer's disease will be essential to support therapeutic development. The authors first surveyed which mouse cells of myeloid origin expressed mouse CD33 and then went on to test the uptake of different cargoes in different mouse and human cell lines and primary microglia, with the goal to compare the relative effects of the mouse and human CD33 proteins. They also sought to establish whether cell surface expression in macrophages depended on the adaptor protein DAP12, although the purpose of this wasn't fully explained.

Overall impression of the work

As the main motivation was to compare the relative function of human and mouse CD33 on phagocytosis, there are grave concerns that this was not achieved by the experimental design and assays utilised in this study.

Specific comments, with recommendations for addressing each comment

Figure 2 - The use of a macrophage cell line (RAW264.7) to assess the impact of the presence/absence of Dap 12 on the expression of CD33 (results section "Dap12-dependent expression of mCD33" and Figure 2) is not optimal (even though practically convenient) in light of the negligible expression of endogenous CD33 found in macrophages in Figure 1 (results section "Expression analysis of mCD33 on immune cell subsets"). The data presented in Figure 2 WT also appears to support the very low expression of CD33 in these cells. The authors need to choose a more appropriate mouse cell type expressing CD33 for these experiments and ensure that it is directly comparable to experiments on hCD33.

It is not clear what the motivation for the experiment presented in results section "Dap12-dependent expression of mCD33" is, given that Dap12 is widely expressed in these cells under normal endogenous circumstances. It is therefore not surprising that adding Dap12 to WT cells has negligible impact on CD33 levels. Showing an impact on downstream activity (e.g. Syk) would have been a better way to demonstrate the functional relationship and dependence between Dap12 and CD33.

Figure 3 - Using the macrophage cell line (RAW264.7) which has negligible expression of CD33 also undermines the investigation of the impact of CD33 on phagocytosis (results section "Loss of mCD33 expression does not alter phagocytosis in RAW254.7 cells", Figure 3). If these cells don't normally utilise CD33, then the findings in Figure 3 of an absence of uptake, regardless of presence/absence of CD33 is not very surprising. The authors need to choose a more appropriate mouse cell type expressing CD33 for these experiments and ensure that it is directly comparable to experiments on hCD33.

Furthermore, there may be additional technical explanations for the results from the macrophage (RAW254.7) (results section "Loss of mCD33 expression does not alter phagocytosis in RAW254.7 cells", Figure 3) and primary microglia experiments (results section "Loss of mCD33 expression does not affect phagocytosis in primary mouse microglia", Figure 4). Details were not provided to assess

levels of phagocytosis in the wild-type cells – if the rate of phagocytosis is already at maximal levels, expecting a further increase in the absence of CD33 might not occur in these cells if they have already achieved maximal levels of phagocytosis. Further details need to be provided on the basal levels of uptake and include phagocytic controls (e.g. LPS treatment) to demonstrate the capacity of the cells to increase phagocytosis if stimulated to do so (e.g. by increased CD33).

The authors then compare the results in mouse cells to those generated in human monocyte cell lines THP-1 and U937. But as shown in Figure 1, the monocyte cells are expected to have higher endogenous levels of CD33 expression, compared to the macrophage cells used for the mouse experiments (results section “Enhanced phagocytosis in monocytic cells that do not express hCD33, Figure 5). These experiments need to be repeated using ‘equivalent’ cells types from mouse and human to allow a direct comparison of mouse and human CD33 to be made.

Results section “Transgene expression of hCD33M in mouse microglia dampens phagocytosis”, Figure 6). Caution needs to be exercised in interpreting the results comparing the species effects from over-expression of a hCD33M transgene with the much more weakly expressed endogenous mCD33 in microglia from the same animal. i.e. The hCD33M transgene contains the CAG promoter which is a strong synthetic promoter frequently used to drive high levels of gene expression in mammalian expression vectors. Therefore, strong conclusions about the relative contribution of mCD33 compared to hCD33 is not possible from such imbalanced expression – Line 246, sentence “these results provide strong support.....”. A different design involving equivalent promoters to drive expression of mouse and human CD33 to enable direct comparison of their relative potency in phagocytosis needs to be carried out. Evidence for equivalent levels of expression (at RNA and protein levels) should also be included to demonstrate that this has been achieved.

Minor amendments

Line 111-112 – For clarity, the unique identifier of the reported functional SNP associated with AD needs to be included – i.e. rs12459419. This will clarify the distinction between this SNP which affects exon 2 splicing and creates an isoform with/without the extracellular sialic acid domain from the ‘tag’ SNP first associated with AD i.e. rs3865444 (e.g. Hollingworth et al, 2011). Both SNPs are in perfect Linkage Disequilibrium.

Line 116-124 – Add details indicating whether this polymorphism is conserved in mouse. i.e. do different mouse strains express mCD33M and mCD33m?

Figure 3 – Details and justification needs to be added for why data from certain clones/cargo combinations were not presented. The text indicates there were 6 clonal lines for CD33^{-/-} and 10 clonal lines for CD33^{+/+}. Data for some of these clones has been excluded from some of the graphs. e.g. Figure 3B (TRITC) appears to include data from N=6 CD33^{-/-} and N=9 CD33^{+/+} while Figure 3C (Pacific blue) appears to include data from N=5 CD33^{-/-} and N=6 CD33^{+/+}.

Reviewers' comments:

Reviewer #1 (Remarks to the Author):

Bhattercherjee et al. showed expression of murine CD33 (mCD33) on neutrophils and microglia. Furthermore, Bhattercherjee et al. demonstrated that mCD33 surface expression is dependent on Dap12. Bhattercherjee et al. also observed that deletion of human CD33 (hCD33), but not mCD33, increased phagocytosis in several in vitro models and that transgenic hCD33 expression in mice resulted in a decrease of phagocytosis.

The manuscript is well-written and discussed. Data are of high relevance for understanding the differences between murine and human CD33 signaling.

Response – Thank you for the positive feedback on the manuscript.

However, there are some points that should be addressed more carefully:

- The histograms with multiple overlays (Fig. 3-6) are hard to distinguish. The use of colors and/or filled histograms would make it easier to distinguish. Further, in Fig. 6 (B-E and F-I) the labeling related to transgenic hCD33M expression in mice (WT or mCD33^{-/-} background) is missing.*

Response – We agree that color would help better distinguish the conditions. Therefore, we have added color to all the graphs and scatter plots for Figures 2-6 to make things clearer. We have also added the background (mCD33^{+/+} or mCD33^{-/-}) in Figure 6, as suggested.

- The subtitles in the results part indicating increased phagocytosis are partly misleading since small size particles such as dextran are taken up via fluid-phase endocytosis or clathrin-dependent endocytosis and thus should not be count to phagocytosis. Moreover, Bhattercherjee et al. did not exclude the possibility of dextran and Aβ to simple bind to the cells without getting internalized.*

Response – We agree with the reviewer that broadly describing the ability of CD33 to regulate uptake of cargo as being only phagocytosis in the subtitles may not be representative. Therefore, we have changed the subtitles to replace 'phagocytosis' with 'cargo uptake' in these subtitles. With regard to the second point about internalization, I would kindly ask the reviewer to look at our response to inquiry #4 to the Editor (see above) on three additional experiments/controls that we have included within our revised manuscript to address this issue.

- The competitive phagocytosis assay in adult primary CD45.2 WT and mCD33^{-/-} versus CD45.1 is unclear. If mCD33 expression in CD45.2 mice is so little that a knockout does not give reliable results in phagocytosis assays, the choice of the mouse line/cells or the readout is questionable. A more sensitive readout system would be better to detect small changes.*

Response – We agree that the low expression of mCD33 in primary adult mouse microglia makes it difficult to definitely conclude that mCD33 cannot regulate phagocytosis. As outlined in our response to inquiry #3B to the editor (see above), three additional experiments were carried out to precisely address the point about low expression of mCD33. Briefly, these three cell types are BV-2 cells, polarized neonatal microglia, and lentiviral overexpression of mCD33 in U937 cells. The conclusions of these studies is that even with higher levels of mCD33 expression, we still find no evidence that mCD33 represses cargo uptake.

• *The statistics of the relative phagocytosis (Fig. 3 H, Fig. 6) appears to be incorrect as the control is always set to 100 without deviation from its mean due to normalization. This increases type I errors in statistics. It would be better to take either raw data or to normalize each value of both, control and experimental group, to the mean of the control group.*

Response – We sincerely thank the reviewer for pointing out this oversight on our part. As suggested by the reviewer, we have normalized to the mean of the control group in these experiments to 100%. Overall, there were very minor changes in the level of statistical significance once the data was replotted in this way, but the overall conclusions do not change that mCD33 does not repress cargo update whereas hCD33 does.

Reviewer #2 (Remarks to the Author):

This study by Bhattecherjee et al describes the role human CD33 in phagocytosis, with an emphasis on microglia and abeta clearance in relation to Alzheimer’s disease (AD). A major claim is that mice expressing human CD33 in the microglial lineage provide a new and valuable model for AD research. However, the authors do not show this. The least that would be necessary, and very interesting indeed given the fact that abeta in mice does not spontaneously aggregate, is to show that hCD33M expression in microglia in APP/PS1 mice reduces abeta clearance and accelerates plaque deposition. It is crucial to demonstrate this in order to substantiate the claim, in particular also because contradicting findings regarding the role of mCD33 have been published, and all data described are derived from in vitro cellular assays using a single technique.

Response – The three major critics brought up here are: (A) the potential ability to modulate plaque deposition *in vivo*, (B) the contradicting findings for mCD33 compared to a published report, and (C) the use of a single technique and only the use of in vitro assays. Responses to each of these three points are address below:

(A) We agree that crossing our hCD33M transgenic line onto an appropriate mouse model of plaque accumulation will provide to be very informative. These studies are underway but will take significant time to age and evaluate. These studies are beyond the scope of this manuscript since its focus is principally phagocytosis at the cellular level, with the comparison between hCD33 and mCD33. We have updated the discussion to indicate that ongoing experiments in our laboratory aim to address the role of hCD33M in plaque deposition in the future: “these hCD33M transgenic mice should provide a valuable model to test the role of hCD33M in regulating plaque accumulation *in vivo*, which is currently being tested in ongoing studies in our laboratory.”.

(B) We fully acknowledge in our manuscript that our results differ from the one report by Griucic et al. (Neuron, 2013) and in our discussion we suggest a few potential reasons for these differences, including the non-pure C57Bl/6J background that the KO mice are on from the commercial supplier (JAX) as well as the possibility that mCD33 modulates plaque accumulation independent of regulating phagocytosis. It is worth noting that in our original manuscript, we examined primary adult microglia, whereas the cell-based assays used by Griucic *et al.* used microglia from neonatal expanded microglia. As described in response number #3B to the editor (see above), we have additionally carried out studies with this type of microglia. In these cells, we still did not observe any differences between WT and mCD33^{-/-} cells in terms of phagocytosis (**Fig. 4F,G**), whereas suppression due to hCD33M was clearly observable (**Sup. Fig 6**), providing yet another piece of evidence supporting the intrinsic difference between mCD33 and hCD33.

(C) The primary readout of our manuscript is indeed phagocytosis. CD33 could potentially regulate other processes in microglia, but a role in suppressing phagocytosis is the primary and only function that has been examined by others. As also brought up by reviewer #1, key controls were missing to conclusively demonstrate that our flow data represents internalization of cargo. Therefore, controls with Cytochalasin-D (**Fig. 3J,K** and **Fig. 5L**) and time-dependent and temperature-dependent assays (**Sup. Fig 3 and 4**) were carried to ensure that we were observing an energy-dependent endocytic mechanism that is in line with phagocytosis. Furthermore, we have employed imaging flow cytometry (**Fig. 5K,L**) to more directly monitor uptake of polystyrene beads; this technique has the advantages of being able to easily analyze large numbers of cells and in competition, like traditional flow cytometry, but has the added benefit that beads stuck to the surface of the cell can be differentiated from cells with an internalized bead.

I also have some concerns regarding the data. It may be my ignorance in the field of flow cytometry, but in all panels where polystyrene phagocytosis is measured, I don't see this back in the flow cytometry charts, which look very different from the other three cargos. This should be explained in more detail. Also, I don't see solid grey (or blue?) lines in any of the charts, whereas the legends indicate that there should be.

Response – There are two differences between the polystyrene beads and the other three types of cargo used in our studies (dextran, aggregated A β ₁₋₄₂, and myelin), which both relate to the large size of these beads. The first is that only a percentage of the cells take up the beads, while for the other cargo all the cells are taking up different amounts of the cargo. The second is

that the polystyrene beads are much brighter, therefore, uptake of even one bead gives increased in fluorescence in the Pacific Blue channel. Therefore, for the polystyrene beads, the cells on the left-hand side of the plot represent those that have not taken up beads. If cells not incubated with the beads were overlaid, they would be directly under the peak on the left, which would make it crowded and non-distinguishable with 4 sets of lines directly on top of each other. Indeed, these differences are reflected in how we report uptake with the polystyrene beads as '% Pacific Blue positive' whereas for the other three cargo it is purely the MFI of the respective fluorophores. In the figure legend of Figure 3, we have added a comment to clarify that cells without polystyrene beads are not shown for clarity.

Reviewer #3 (Remarks to the Author):

Brief summary of the manuscript

The authors were interested in understanding whether the CD33 mouse and human receptor orthologs which differ at two positions in the protein, have the same impact on uptake of different cargoes, including A β species. This is of interest because a DNA variant in CD33 is associated with AD risk and establishing the best CD33 AD mouse model able to capture the functional pathway of CD33 and relevant to Alzheimer's disease will be essential to support therapeutic development. The authors first surveyed which mouse cells of myeloid origin expressed mouse CD33 and then went on to test the uptake of different cargoes in different mouse and human cell lines and primary microglia, with the goal to compare the relative effects of the mouse and human CD33 proteins. They also sought to establish whether cell surface expression in macrophages depended on the adaptor protein DAP12, although the purpose of this wasn't fully explained.

Overall impression of the work

As the main motivation was to compare the relative function of human and mouse CD33 on phagocytosis, there are grave concerns that this was not achieved by the experimental design and assays utilised in this study.

Specific comments, with recommendations for addressing each comment

Figure 2 - The use of a macrophage cell line (RAW264.7) to assess the impact of the presence/absence of Dap 12 on the expression of CD33 (results section "Dap12-dependent expression of mCD33" and Figure 2) is not optimal (even though practically convenient) in light of the negligible expression of endogenous CD33 found in macrophages in Figure 1 (results section "Expression analysis of mCD33 on immune cell subsets"). The data presented in Figure 2 WT also appears to support the very low expression of CD33 in these cells. The authors need to choose a more appropriate mouse cell type expressing CD33 for these experiments and ensure that it is directly comparable to experiments on hCD33.

Response – We agreed with the reviewer that to fully rule out that mCD33 cannot repress phagocytosis it was better to test it under conditions where mCD33 is more robustly expressed. Accordingly, we carried out three additional studies in BV-2 cells, neonatal microglia polarized to induce higher mCD33 expression, as well as lentiviral transduction of mCD33 in U937 cells. I would kindly ask the reviewer to read our response to inquiry #3B from the editor for a full recap of these three experiments.

It is not clear what the motivation for the experiment presented in results section "Dap12-dependent expression of mCD33" is, given that Dap12 is widely expressed in these cells under normal endogenous circumstances. It is therefore not surprising that adding Dap12 to WT cells has negligible impact on CD33 levels. Showing an impact on downstream activity (e.g. Syk)

would have been a better way to demonstrate the functional relationship and dependence between Dap12 and CD33.

Response – Demonstrating that mCD33 requires Dap12 for expression on the cell surface is something that had never been demonstrated previously, so it is novel. It is a similar finding to other Siglecs that have a key transmembrane lysine residue, so not entirely surprising but the point is that it had never been formally demonstrated. The point of the Dap12 transfection was to genetically complement the Dap12^{-/-} cells. This complementation indeed showed that reintroduction of Dap12 restored mCD33 cell surface expression levels. As detailed in our response to inquiry #5 by the editor, we have gone one step further by deleting Dap12 in U937 cells and showed that hCD33 expression is not perturbed, emphasizing this feature is unique to mCD33.

Figure 3 - Using the macrophage cell line (RAW264.7) which has negligible expression of CD33 also undermines the investigation of the impact of CD33 on phagocytosis (results section “Loss of mCD33 expression does not alter phagocytosis in RAW254.7 cells”, Figure 3). If these cells don't normally utilise CD33, then the findings in Figure 3 of an absence of uptake, regardless of presence/absence of CD33 is not very surprising. The authors need to choose a more appropriate mouse cell type expressing CD33 for these experiments and ensure that it is directly comparable to experiments on hCD33.

Response – As detailed in our response to this reviewer's first inquiry, we have examined three additional cell types that express significantly more mCD33 than RAW264.7 cells. In these new experiments - where mCD33 is expressed at higher levels - we still find no evidence that mCD33 represses phagocytosis.

Furthermore, there may be additional technical explanations for the results from the macrophage (RAW254.7) (results section “Loss of mCD33 expression does not alter phagocytosis in RAW254.7 cells”, Figure 3) and primary microglia experiments (results section “Loss of mCD33 expression does not affect phagocytosis in primary mouse microglia”, Figure 4). Details were not provided to assess levels of phagocytosis in the wild-type cells – if the rate of phagocytosis is already at maximal levels, expecting a further increase in the absence of CD33 might not occur in these cells if they have already achieved maximal levels of phagocytosis. Further details need to be provided on the basal levels of uptake and include phagocytic controls (e.g. LPS treatment) to demonstrate the capacity of the cells to increase phagocytosis if stimulated to do so (e.g. by increased CD33).

Response – This is an excellent point and we agree that is critical to show that experiments are being carried out at a time point where phagocytosis is not saturated. Prior to carrying out our original studies, we had indeed looked at a time dependency of phagocytosis and shown that it was linear up to 30 minutes. To demonstrate this more clearly with WT and KO cells, we have repeated phagocytosis in WT and KO RAW264.7 cells (**Sup. Fig 3**), as well as U937 cells (**Sup. Fig 4**) over 80 minutes. The results of these time courses demonstrate that the 30 minute timepoint used in all our experiments is not saturating. More importantly, effects (or lack thereof) stemming from loss of CD33 hold up from 10 to 80 minutes.

The authors then compare the results in mouse cells to those generated in human monocyte cell lines THP-1 and U937. But as shown in Figure 1, the monocyte cells are expected to have higher endogenous levels of CD33 expression, compared to the macrophage cells used for the mouse experiments (results section “Enhanced phagocytosis in monocytic cells that do not express hCD33, Figure 5). These experiments need to be repeated using ‘equivalent’ cells

types from mouse and human to allow a direct comparison of mouse and human CD33 to be made.

Response – To more fairly compare between mCD33 and hCD33 we stably transduced mCD33 or hCD33M into hCD33^{-/-} U937 cells. In our revised manuscript, we show that transduction of hCD33M clearly repressed phagocytosis (**Fig. 5M,N**). Using the same approach to express mCD33 led to significant expression of mCD33 (**Sup Fig. 6A**), but not to the same as hCD33M. On the other hand, a K252A mutant of mCD33, where the critical transmembrane lysine is mutated, leads to very high expression levels of mCD33. Importantly, neither WT nor K252A mCD33 repressed phagocytosis (**Sup Fig. 6B**). In the discussion of the revised manuscript, we have discussed the significance of the inability of high levels of the mCD33 K252A mutant to repress phagocytosis: “The inability of mCD33 to repress phagocytosis may be related to the absence of an ITIM since overexpression of a K252A mutant of mCD33 achieved a high level of expression in hCD33^{-/-} U937 cells yet still did not repress phagocytosis.”

Results section “Transgene expression of hCD33M in mouse microglia dampens phagocytosis”, Figure 6). Caution needs to be exercised in interpreting the results comparing the species effects from over-expression of a hCD33M transgene with the much more weakly expressed endogenous mCD33 in microglia from the same animal. i.e. The hCD33M transgene contains the CAG promoter which is a strong synthetic promoter frequently used to drive high levels of gene expression in mammalian expression vectors. Therefore, strong conclusions about the relative contribution of mCD33 compared to hCD33 is not possible from such imbalanced expression – Line 246, sentence “these results provide strong support.....”. A different design involving equivalent promoters to drive expression of mouse and human CD33 to enable direct comparison of their relative potency in phagocytosis needs to be carried out. Evidence for equivalent levels of expression (at RNA and protein levels) should also be included to demonstrate that this has been achieved.

Response – See our response to the last inquiry. Both mCD33 and hCD33M were virally transduced into U937 cells with the same promoter but only hCD33M repressed phagocytosis.

Minor amendments

Line 111-112 – For clarity, the unique identifier of the reported functional SNP associated with AD needs to be included – i.e. rs12459419. This will clarify the distinction between this SNP which affects exon 2 splicing and creates an isoform with/without the extracellular sialic acid domain from the ‘tag’ SNP first associated with AD i.e. rs3865444 (e.g. Hollingworth et al, 2011). Both SNPs are in perfect Linkage Disequilibrium.

Response – Thank you for this good suggestion. In the introduction of our revised manuscript, we have added the identifier that affects splicing within Exon 2.

Line 116-124 – Add details indicating whether this polymorphism is conserved in mouse. i.e. do different mouse strains express mCD33M and mCD33m?

Response – There is no evidence for the presence of an equivalent shorter isoform of CD33 in mouse. We have added a sentence indicating this in the introduction, as well as a reference to back-up this statement: “It is noteworthy that the hCD33m isoform, lacking its glycan-binding domain, appears to be unique to humans²⁸.”

Figure 3 – Details and justification needs to be added for why data from certain clones/cargo combinations were not presented. The text indicates there were 6 clonal lines for CD33^{-/-} and

10 clonal lines for CD33^{+/+}. Data for some of these clones has been excluded from some of the graphs. e.g. Figure 3B (TRITC) appears to include data from N=6 CD33^{-/-} and N=9 CD33^{+/+} while Figure 3C (Pacific blue) appears to include data from N=5 CD33^{-/-} and N=6 CD33^{+/+}.

Response – Thank you for pointing out these two discrepancies. We did initially isolate ten clones of CD33^{-/-} and 6 of CD33^{+/+}. However, one of these KO clones ended up expression intermediate levels of mCD33 so we ended up not using it. Accordingly, the number of clones used has been updated. In Figure 3C of our original manuscript (Figure 3D in the revised manuscript) we indeed reported N=5 CD33^{-/-} and N=6 CD33^{+/+} as pointed out by the reviewer. This was from one of our initial datasets when all the clones were not ready for phagocytic assays and it was an oversight on our part to include this dataset. As all these experiments were repeated 2-3 times, we have another dataset for the polystyrene beads in RAW264.7 cells in which all 6 WT and 9 KO clones were used. We have included this more complete dataset in our revised manuscript. For the microscopy assay, since these experiments were more experimentally and computationally intensive than the flow-based assay, we randomly chose 4 mCD33^{-/-} clones to compare to the 8 mCD33^{+/+} clones. To be more clear about this last point, we have updated the figure legend and methods to account for the number of clones used in this microscopy assay.

REVIEWERS' COMMENTS:

Reviewer #1 (Remarks to the Author):

All comments raised by me were adequately addressed in the revised manuscript

Reviewer #2 (Remarks to the Author):

This revised manuscript offers a great improvement compared with the original submission. Many new experiments have been added, unclarities are now explained in more detail and the changed color schemes for the figures allows for much easier interpretation of the data.

As for my specific comments, they have all been addressed. I appreciate the fact that experiments in AD mice are ongoing and may be beyond the scope of the present paper. Also, the cautionary note the authors included on this topic in the discussion is much appreciated. As a final remark I would like to suggest to also change the title as to not suggest the presentation of a mouse model for AD. It would be more appropriate in my view to stick to the data (mCD33 and hCD33 have different phagocytic properties) and the implications these might have (potential implications for amyloid clearance in Alzheimer's disease), and come up with a creative and attractive title along these terms.